# Effects of Vertical Fiscal Imbalance on Fiscal Health Expenditure Efficiency—Evidence from China

**DOI:** 10.3390/ijerph20032060

**Published:** 2023-01-23

**Authors:** Simin Zhang, Zhikai Wang

**Affiliations:** School of Economics, Zhejiang University, Hangzhou 310027, China

**Keywords:** vertical fiscal imbalance, fiscal health expenditure efficiency, transfer payment

## Abstract

Improving fiscal health expenditure efficiency is an inherent requirement of the strategy of “healthy China” and “high-quality development”. The outbreak of COVID-19 has highlighted the importance of efficient health system. First, this paper systematically sorts out the multiple theoretical mechanisms of the positive and negative relationship between vertical fiscal imbalance and fiscal health expenditure efficiency. Secondly, a comprehensive index system, including the quantity and quality of medical services, is constructed, and the super-efficiency DEA model is used to measure fiscal health expenditure efficiency. There are obvious differences between eastern and western regions. Finally, the fixed effect model is constructed to carry out empirical research and it is found that the vertical fiscal imbalance in China has an overall positive and significant impact on the fiscal health expenditure efficiency, which is mainly achieved by optimizing the resources allocation between primary medical institutions and hospitals. Heterogeneity analysis shows that transfer payment scale has a corrective effect on the vertical fiscal imbalance’s effect. The result of quantile regression shows that the impact of vertical fiscal imbalance is not constant, and it gradually turns from positive to negative along with the improvement of fiscal health expenditure efficiency.

## 1. Introduction

Medical and health service is an important part of the government’s public services, and it is also a major livelihood undertaking related to the vital interests of every family and every public. In order to improve the accessibility of residents’ medical and health services, the Chinese government continues to increase financial support. The new medical reform plan in 2009 clearly pointed out that the government should establish the dominant position in the field of health financing and propose an additional 850 billion financial funds within three years to support the construction of medical and health services. Under the continuous promotion of governments at all levels, the scale of government health expenditure keeps increasing, which alleviates the defect of insufficient health resources to a certain extent. In 2019, the Chinese government’s health expenditure was 1666.534 billion yuan, accounting for 7% of the government’s financial expenditure. However, the uneven allocation of medical resources leads to the difficulty and high cost of medical treatment, the limited improvement of residents’ medical utilization, and the low efficiency of fiscal health expenditure. Under the background of “healthy China” and “high-quality development” strategy, improving the residents’ medical utilization and fiscal health expenditure efficiency have become the key of China’s health reform. In particular, the outbreak of COVID-19 has highlighted the importance of an efficient health system. It is of great significance to analyze the influencing factors of fiscal health expenditure efficiency and to realize the rational allocation of health resources for optimizing medical and health services, and improving people’s livelihood undertakings.

The fiscal decentralization system has a profound impact on local government behavior, mainly reflected in fiscal revenue and expenditure behaviors of local government such as tax competition [1,2,3,4], land transfer [5,6,7], distortion of public expenditure structure [8], as well as a series of economic and social chain reactions. Based on the dominance of the Chinese government in the field of health financing, the fiscal decentralization system is bound to have an impact on the fiscal health expenditure efficiency. At present, there exists considerable literature on this based on the perspective of decentralization. However, the measurement of fiscal decentralization indicators is complicated, including fiscal revenue indicators, fiscal decentralization indicators and fiscal self-sufficiency, which imply different mechanisms and may have opposite effects on the fiscal health expenditure efficiency [9,10,11,12]. Since tax distribution reform in 1994, the fiscal power of local governments has been greatly weakened, but their expenditure responsibilities have been strengthened rather than weakened, resulting in an increase in the gap between local governments’ own revenue and expenditure. Namely. Additionally, the problem of vertical fiscal imbalance has become increasingly prominent. This paper attempts to separately study the impact of vertical fiscal imbalance on the fiscal health expenditure efficiency, which is helpful to more clearly analyze and understand the effect of the fiscal decentralization system on medical care in China.

Compared to previous studies, the possible marginal contribution of this paper is mainly reflected in the following three aspects. First, due to the complex mechanism implied by different indicators of fiscal decentralization, the conclusions of the literature are different. This paper individually selects the vertical fiscal imbalance as the core explanatory variable to conduct a more detailed and clear empirical analysis, and to clarify the specific mechanism from the perspective of the scale and structure of health resources allocation. It reveals the behavioral characteristics of local governments in the field of health in China, and further expands related research on Chinese-style fiscal decentralization. It is concluded that the vertical fiscal imbalance has a significant and positive influence on fiscal health expenditure efficiency, which is mainly achieved by narrowing the gap of medical utilization between hospitals and primary medical institutions. Second, most literature only considers the quantity index but ignore the quality index when calculating efficiency. The output indicators in this paper comprehensively consider the quantity and quality, hoping to more comprehensively reflect the resident medical utilization. Moreover, the further robustness analysis shows that the fiscal health expenditure efficiency is positively and significantly correlated with the self-rated health indicators and has no significant correlation with the total out-of-pocket expenses of residents. Therefore, it is further confirmed that the fiscal health expenditure efficiency calculated in this paper can truly reflect the quality of health care without the increase in residents’ medical burden. Thirdly, this paper further analyzes the heterogeneity from the perspective of transfer payment system, so as to comprehensively and profoundly understand the behavior of local governments in the field of health under the fiscal decentralization system. The conclusion is that the transfer payment mainly plays a corrective role in vertical fiscal imbalance.

The follow-up arrangement of the article is as follows: The second part summarizes the relevant literature; the third part constructs the theoretical mechanism of the effect of vertical fiscal imbalance on the fiscal health expenditure efficiency; the fourth part introduces the measurement method and index selection of fiscal health expenditure efficiency in detail; the fifth part introduces the model setting, variable definition, data sources and statistical characteristics; the sixth part includes the baseline regression results, robustness test and heterogeneity analysis, and empirically tests the ways to improve the fiscal health expenditure efficiency; the last part is the conclusion and policy recommendations.

## 2. Institution Background and Literature Review

### 2.1. Tax-Sharing Reform and Transfer Payment System

Since the tax-sharing reform in 1994, the financial relations between the central and local governments have undergone major adjustments. The fiscal power has gradually moved up, but the local government’s responsibilities have not been adjusted accordingly, leading to a widening gap of fiscal revenue and expenditure, namely vertical fiscal imbalance. The financial pressure on local governments has also increased, with important implications for their behavior. With the relationship adjustment between central and local governments, the central government has increased the intensity of transfer payments to local governments, mainly in the form of tax rebates, general transfer payments and special transfer payments. The tax rebates were incorporated into general transfers in 2019. General transfer payments are mainly used to bridge the financial gap between different regions, and local governments have the autonomy to allocate funds. It mainly adopts factor method for formula allocation, including economic, social and other factors. The formula method refers to the calculation of the difference between the standard fiscal revenue and the standard fiscal expenditure multiplied by the transfer payment coefficient. The fund allocation method of general transfer payment is relatively objective and scientific, and local governments have less room for rent-seeking [13]. However, the freedom control of general transfer payment provides conditions for the local government’s fiscal expenditure bias of “emphasizing production and neglecting people’s livelihood” to some extent [14]. For special transfer payments, it specifies the use of funds and effectively urges local governments to achieve specific national economic and social goals, usually requiring local governments to provide corresponding matching funds [15]. It comprehensively adopts the factor method and the project method to allocate funds. The project method mainly adopts the way of competitive review to allocate funds, which has defects such as low transparency and difficult assessment, resulting in problems such as “hunting Ministries for Money” and overlapping projects [16]. In 2019, the items in the original special transfer payments that clearly belonged to the common financial authority of the central and local governments, were incorporated into the general transfer payments and established as the common financial authority transfer payments. This means a major step forward in the division of fiscal powers and expenditure responsibilities between central and local governments. As mentioned above, fiscal decentralization system has an important influence on government’s public expenditure behavior. This paper focuses on the influence of vertical fiscal imbalance on fiscal health expenditure efficiency.

### 2.2. Related Literature on Fiscal Health Expenditure Efficiency

Basic medical and health services include inpatient services, outpatient services and public health services. This paper focuses on the improvement of residents’ medical utilization, including effective utilization of medical services, improvement of health level, and reduction of unnecessary medical burden. However, the data at the provincial level in China lack relevant indicators such as mortality rate for measurement, so this paper tries to use the index of fiscal health expenditure efficiency for measurement. Regarding this index, the indicators of health input and output in the literature are not uniform. One type of literature directly measures the efficiency of fiscal expenditure as fiscal health expenditure efficiency, taking government health expenditure as the input index and health resources such as the number of health institutions, number of health technicians and number of beds as output index [10,17]. Sometimes, they take government health expenditure as input index, and medical resources such as the number of health institutions and medical utilization such as outpatient visits as the output indicator [18,19,20,21]. Another type of literature measures the utilization efficiency of health resources as fiscal health expenditure efficiency, taking the number of health institutions, the number of health technicians and the number of beds as input indicators, and medical utilization such as outpatient visits, hospital admissions and discharge times, or health outcomes such as life expectancy, infant mortality rate and child immunization rate as output indicators [22,23]. This paper chooses the utilization efficiency of health resources as fiscal health expenditure efficiency, which can more directly reflect the medical utilization of residents. However, little literature takes medical services quality as output indicators, which is exactly the purpose of the government’s provision of medical and health services and can more truly reflect the medical utilization of residents. There are only a few literature efforts in China that uses case fatality rate in the observation room, success rate of rescue for critically ill patients, inpatient fatality rate and emergency fatality rate as quality characterization indexes [24,25]. When describing the output index of fiscal health expenditure efficiency, this paper comprehensively selects the quantity and quality of medical services, especially the prenatal examination rate and incidence rate of class A and B infectious diseases, which reflect the utilization of public health services.

### 2.3. Literature on Vertical Fiscal Imbalance

Scholars generally believe that vertical fiscal imbalance originates from the inherent arrangement of the decentralization system. Vertical fiscal imbalances naturally arise when countries decentralize financial authority without maintaining equal fiscal decentralization of property rights [26,27]. As there is no universally accepted definition of vertical fiscal imbalance in the academic field, the measurement methods vary greatly. At present, empirical studies generally measure vertical fiscal imbalance based on the definition of traditional methods, that is, the degree of mismatch between local government revenue and expenditure [28]. The most commonly used measurement method is the ratio of independent source income to self-owned expenditure, but there are various views on the definition of independent source income. One is that independent sources only include local governments’ own revenue. The second argument is that independent income should also include shared taxes, which can also be spent freely by local governments [29]. The third argument is that independent revenue should include all the revenue that can be freely spent by local governments, including unconditional transfer payments, etc. Based on the definition of traditional methods, domestic literature generally adopts the measurement method by Eyraud and Lusinyan (2013), and fully considers the revenue and expenditure decentralization [30]. Jia Junxue et al. measured the vertical fiscal imbalance as 1—(total local income/total local expenditure) [31]. Li Yongyou et al. defined self-owned expenditure as local budget expenditure excluding local central expenditure and special transfer payment financing expenditure, while self-owned income was defined as local budget income excluding shared tax [32]. In addition to the general budget, it also includes the fund budget and extrabudgetary revenue and expenditure, which is a more comprehensive measure of vertical financial imbalance. Further calculation shows that the correlation coefficient between the index in this paper and the results of the previous two measurement methods is 0.98 and 0.95. Due to the difficulty of data acquisition, we select the calculation methods of Eyraud and Lusinyan (2013), which not only have a high correlation with the indicators measured by Li Yongyou, but can also well reflect the fiscal decentralization system of China.

A large number of literature efforts have studied the effects of vertical fiscal imbalance on fiscal health expenditure efficiency, and the research conclusions are mainly divided into two levels. On the one hand, moderate vertical fiscal imbalance is inevitable and reasonable, and is conducive to optimizing the structure of local government’s public expenditure and improving government expenditure efficiency. The central government corrects the financial expenditure behavior of local governments by designating the use of funds allocated to them [33]. At the same time, the central government can design a reasonable reward and punishment mechanism to achieve policy incentives for local governments and encourage local governments to better perform their duties. However, when the degree of vertical fiscal imbalance is high, it will exacerbate the imbalance of the local government’s public expenditure structure and reduce the fiscal expenditure efficiency. Excessive vertical fiscal imbalance will further increase the local government’s reliance on transfer payments and exert pressure on the central government, with aggressive fiscal revenue and expenditure measures of local government. Vertical fiscal imbalance has threshold effect on the efficiency of social expenditure in education and public health [34]. Moderate vertical fiscal imbalance will stimulate the enthusiasm of local governments to participate in environmental governance, while excessive vertical imbalance will stimulate local governments to pay more attention to productive expenditures, reduce environmental governance expenditures, and reduce the green development efficiency [35]. When vertical fiscal imbalance exceeds the threshold value, its overall positive effect on economic growth and its institutional dividend decline rapidly [36]. It is found that with the increase of vertical fiscal imbalance, its positive effect on resource allocation efficiency gradually recedes [37]. In general, although vertical fiscal imbalance violates the efficiency principle of consistent power and responsibility and excessive vertical fiscal imbalance has many adverse effects, but moderate vertical fiscal imbalance has the dual role of improving efficiency and ensuring fairness.

However, few articles directly study the effects of vertical fiscal imbalance on medical care, mainly focusing on the effects of fiscal decentralization. Zhu Deyun et al. found that fiscal expenditure decentralization had a significant positive effect on fiscal health expenditure efficiency, while fiscal revenue decentralization and fiscal autonomy had a negative effect on it, showing nonlinear characteristics [38].Cui Zhikun et al. adopted the data of Jiangsu Province from 2009 to 2015 and also found that expenditure decentralization improved the fiscal health expenditure efficiency, while income decentralization significantly hindered the fiscal health expenditure efficiency [39].The reason may be that local governments excessively pursue political achievements, and the strengthening of fiscal revenue decentralization will make government expenditure biased towards the infrastructure construction. Meanwhile, the strengthening of expenditure decentralization will take advantage of information from local government, which can avoid the mismatch of public goods such as medical and health services. Additionally, the existence of government competition also enables local governments to improve public services to attract residents. In addition, some studies have found that both revenue decentralization and expenditure decentralization significantly increased local governments’ public health expenditures and medical resources such as beds and medical personnel, but there are also non-linear effects [40].The vertical fiscal imbalance in this paper comprehensively measures the degree of revenue decentralization and expenditure decentralization, and vertical fiscal imbalance is directly proportional to the former and inversely proportional to the latter. Additionally, it can more comprehensively reflect the influence of the current fiscal decentralization system.

### 2.4. Related Literature on Transfer Payment System

The above literature also shows that the level of transfer payment plays an important role in regulating the effect of vertical fiscal imbalance, including the effect on the medical field. On the one hand, in terms of the scale of public expenditure, a large number of studies have shown that transfer payments have “flypaper effect” and “public pool effect” [41,42,43], thus reducing the efficiency of expenditure. “Flypaper effect” means that, compared to the local government’s own fiscal revenue, the same amount of unconditional transfer payment more stimulates the local government’s spending expansion motivation, resulting in many adverse effects [44]. This is mainly due to the existence of income effect and price effect, that is, transfer payment is regarded as non-self-owned fiscal revenue and reduces the marginal cost of local government in providing public services [45,46]. The “public pool” effect means that local governments use the public pool provided by transfer payments to share expenditure costs, which leads to their easy neglect of expenditure efficiency [47,48,49].

However, in terms of the structure of public expenditure, the original intention of the transfer payment system is to balance regional financial resources, promote the equalization of public services, achieve the national goals of improving people’s livelihood and guiding healthy economic growth. Due to the design defects in transfer payment system, there are many possibilities for its impact on the structure of public expenditure. The empirical results of some literature show that transfer payments provide conditions for local governments to favor production expenditures, and even special transfer payments are limited in guiding local governments to improve people’s livelihood due to “soft budget constraints”, inadequate supervision and other reasons [50,51,52], which is not helpful to the people’s livelihood expenditure efficiency, including medical care. The results of some literature show that equalized transfer payment can significantly optimize the structure of public expenditure. In particular, special transfer payments for education have played a role in correcting local government expenditure behavior [53,54,55], thus improving people’s livelihood expenditure efficiency. A literature on Sichuan and Chongqing shows that the influence of transfer payment on the structural bias of citizen public expenditure presents an “inverted U-shaped” relationship [56].

## 3. Discussion of Mechanism

### 3.1. Composition of Fiscal Health Expenditure Efficiency

Scale efficiency and structure efficiency. Haixiang Xiao constructed a fiscal health expenditure efficiency system from three levels: Scale efficiency, structural efficiency, and production efficiency [57]. Additionally, production efficiency is affected by scale efficiency and structural efficiency. As mentioned above, in terms of production efficiency, this paper mainly investigates the utilization efficiency of health resources, which means the improvement of residents’ medical utilization and even the improvement of their health level. Scale efficiency mainly affects production efficiency by exerting scale effect. With the increase of government investment in health resources (such as the number of medical institutions, medical technicians and beds, etc.), the division of labor and coordination effect of public health departments and the ability of health resource planning are improved, and residents’ utilization rate of medical resources is improved, resulting in the increasing return to scale effect, thus improving fiscal health expenditure efficiency. However, when the government’s investment in health resources is too high, it will also disrupt the rational division of labor within the public sector, produce a diminishing return to scale effect, and reduce the fiscal health expenditure efficiency. In terms of structural efficiency, it includes urban and rural allocation, the allocation between medical institutions, and the allocation between regions. This paper mainly studies the allocation of health resources between urban and rural areas. This paper mainly studies the allocation of health resources in hospitals and primary medical institutions. The unbalanced allocation of resources between the two will weaken the health service capacity of primary medical institutions, and residents are more inclined to go to large hospitals for medical treatment, which will exacerbate the problems of difficult and expensive medical treatment. Therefore, there may be redundant input of medical resources in primary medical institutions, and insufficient input of medical resources in hospitals. Furthermore, the medical needs of the population are not met, which is detrimental to their health. Therefore, no matter whether considering the quantity output index or the quality output index, the unbalanced allocation of resources in medical institutions will reduce the overall utilization efficiency of medical resources.

### 3.2. Positive Mechanism of Vertical Fiscal Imbalance on Fiscal Health Expenditure Efficiency

Under moderate vertical fiscal imbalance, the corresponding transfer payment and financial pressure will promote the fiscal health expenditure efficiency. First, financial pressure plays an incentive role. Vertical fiscal imbalance, that is, the mismatch between the local government’s own revenue and expenditure, causes fiscal pressure and urges local governments to take measures of revenue and expenditure to relieve the pressure. However, fiscal expenditure has the characteristics of rigid expansion, and the strategy of reducing expenditure is not feasible. Moreover, in the long run, the measures taken by local governments to expand fiscal revenue, such as tax efforts, land transfer and debt financing, are not conducive to the benign development of local economy. Therefore, in the case of moderate financial pressure, improving the fiscal expenditure efficiency becomes a good choice for local governments to release financial pressure [21,58,59]. Second, transfer payments play a corrective role. The central government often makes up for the fiscal revenue and expenditure gap under the vertical fiscal imbalance through transfer payments. The central government can guide the expenditure behavior of local governments by optimizing the transfer payment system and designating the use of funds, increase investment in medical and other livelihood expenditures, optimize the allocation of medical resources, and thus promote fiscal health expenditure efficiency [47,60]. Similarly, within the moderate range of vertical fiscal imbalance, their decline will lead to the reduction in the fiscal health expenditure efficiency. On the one hand, the effects mentioned above are weaken. On the other hand, as mentioned above, the decline of vertical fiscal imbalance means the rise of revenue decentralization and the decline of expenditure decentralization, which will also encourage the tendency to focus on production expenditure under the official promotion system [61,62]. However, as the country gradually includes medical and other public services in the assessment, local governments should also take into account their performance in public services. Therefore, local governments may have adopted selective response policies in the medical field to achieve the dual goals of economic and livelihood assessment. Obviously, in the current reality, public hospitals can create more revenue and achievements. Therefore, local governments may be more inclined to shift medical resources to large hospitals, which can not only meet the assessment of people’s livelihood, but also generate income for local governments. The large-scale expansion of public hospitals and the relatively low survival status of primary medical institutions are good evidence. Local governments are the owners of public hospitals and primary medical institutions and play an important role in the allocation of talents, beds, equipment and other resources. For example, special talents preferential policies are introduced, land planning and financial subsidies are provided for the scale expansion of public hospitals, and public hospitals are allowed to build buildings [63]. However, such expenditure bias will also aggravate the imbalance of health resources in medical institutions, thus reducing the fiscal health expenditure efficiency.

### 3.3. Negative Mechanism of Vertical Fiscal Imbalance on Fiscal Health Expenditure Efficiency

Excessive vertical fiscal imbalance will exacerbate fiscal pressure and the distorting effect of transfer payments on fiscal expenditure structure, which is not conducive to the improvement of fiscal health expenditure efficiency. On the one hand, excessive fiscal vertical imbalance will increase the pressure of fiscal revenue and expenditure gap, prompting local governments to choose fiscal revenue increase as the best choice to alleviate fiscal pressure. Especially under the “economic championship” of the “top-down” assessment system, local governments tend to favor basic construction spending that can promote economic development and increase tax sources more quickly, while relatively ignore spending on people’s livelihood such as medical care. Moreover, under financial pressure, because hospitals have higher returns than primary medical institutions, local governments may be more inclined to tilt medical resources to large hospitals, aggravating the imbalance of health resources among different medical institutions, thus reducing the fiscal health expenditure efficiency. On the other hand, local governments under the excessively unbalanced system will be more dependent on transfer payments from the central government, probably resulting in “fly-paper effect” and “public pool problem”, which makes local governments more aggressive in their spending strategies and neglect the efficiency of fund use. As mentioned above, it may also distort the structure of local government spending due to soft budget constraints and the shortcomings of the transfer payment system itself. Research shows that the larger the scale of transfer payment is, the stronger the motivation of local governments to increase the expenditure of capital construction and administrative management fees is [50]. Moreover, the expansion of transfer payments may also exacerbate local governments’ efforts to divert funds to large hospitals. As a result, the scale efficiency and structural efficiency of health resources will also be reduced, which will eventually lead to the reduction of the fiscal health expenditure efficiency.

## 4. Materials and Methods

### 4.1. DEA Model and Index System Construction

This paper uses super-efficiency DEA method, one type of DEA, to calculate fiscal health expenditure efficiency. DEA method is a non-parametric estimation method of linear programming, which is used to evaluate the relative effectiveness of decision units in the research object. Compared to other parameter estimation methods such as SFA, the advantage of this method lies in that it does not need to assume the form of production function and is suitable for efficiency calculation of multi-input and multi-output decision units. Compared with the traditional DEA model, the super-efficiency DEA can also compare and sort the efficiency values of decision-making units, especially when there are multiple decision-making units with an efficiency value of 1 [64]. When calculating the fiscal health expenditure efficiency with DEA in this paper, the efficiency value of many provinces is 1, so the super-efficiency DEA method is adopted.

Compared with the traditional DEA model, super-efficiency DEA firstly excludes the decision-making unit evaluated, and calculates the linear combination of inputs and outputs of all other decision-making units to replace its inputs and outputs. The efficiency value of invalid decision-making unit is unchanged, and the efficiency value of effective decision-making unit is larger than that measured by traditional DEA model, and can be sorted and compared. Regarding fiscal health expenditure efficiency, this paper focuses more on how to effectively increase the output of medical utilization while maintaining the input factors of health resources unchanged, and the output index may have variable returns to scale, so super-efficient DEA model based on the variable scale returns under the output-oriented approach are adopted. Its mathematical form is as follows, where i,r,j represents input variable, output variable and decision making unit respectively, and m,q,n is the number of input variable, output variable and decision making unit. σ is the super efficiency value of decision-making unit k, and $\lambda_{j}$ is the ratio of reconstructing an effective decision-making unit relative to decision making unit k.
(1)$$\begin{array}{l} \max\sigma \\ s.t.\\ \left\{ \begin{array}{l} \overset{n}{\underset{\begin{array}{l} j=1\\ j\neq k \end{array}}{\sum }}\lambda_{j}X_{ij}\leq X_{ik}\\ \overset{n}{\underset{\begin{array}{l} j=1\\ j\neq k \end{array}}{\sum }}\lambda_{j}Y_{rj}\geq \sigma Y_{rk}\\ \overset{n}{\underset{\begin{array}{l} j=1\\ j\neq k \end{array}}{\sum }}\lambda_{j}=1\\ i=1,2,\ldots ,m;r=1,2,\ldots ,q;j=1,2,\ldots ,n(j\neq k) \end{array}\right. \end{array}$$

As mentioned above, the fiscal health expenditure efficiency is adopted in this paper to reflect the residents’ medical utilization. Basic health services include public health services and medical services, and medical services include outpatient services and inpatient services. As shown in Table 1, the output indicators selected in this paper cover all aspects of medical and health services, the rate of prenatal check-up and the incidence of class A and B infectious diseases belong to public health services, and the rest belong to medical services. Meanwhile, as mentioned above, the quality of medical and health services is precisely the purpose of the government’s provision, and it can more truly reflect the fiscal health expenditure efficiency. When describing the output indicators, this paper comprehensively considers the quantity and quality indicators of medical services. For example, emergency fatality rate is a qualitative index, and the rest is a quantitative index. The indicators in this paper are derived from China Health Statistical Yearbook. Due to the limitation of data, the quality indicators of medical utilization can be selected in a limited way. The selected input–output indicators are as follows:

### 4.2. Measurement Results of Fiscal Health Expenditure Efficiency

In this paper, the fiscal health expenditure efficiency in 31 provinces of China from 2009 to 2019 was measured and analyzed by region. The measurement results are shown in Figure 1. The average value of fiscal health expenditure efficiency is above 1 in China as a whole, showing a trend of decline, rise and then decline. Additionally, there are great differences among regions. Government health spending is most efficient in the east, hovering around 1.2. The fiscal health expenditure efficiency of western government is the second highest, maintaining above 1. Middle fiscal health spending is the least efficient, at less than 1.

## 5. Model Setting, Variable Definition, Data Source and Statistical Characteristics

### 5.1. Two-Way Fixed Effects Model

In the second stage, considerable literature has shown that the maximum value of traditional DEA efficiency is one, which belongs to the restricted dependent variable. Therefore, Tobit model should be adopted to analyze the influence of various factors on efficiency value. However, it is argued in the literature that efficiency values belong to fractional data rather than being generated by the review process, and that Tobit regression analysis can produce biased and inconsistent estimators [65,66]. Moreover, the maximum value of super-efficiency is not limited to one, and there is no case of truncation. Therefore, in this paper, the ordinary least squares method is used for regression analysis. Among the selection of mixed OLS model, random effects model and fixed effects model, the fixed effects model is finally selected after a series of F tests, LM tests and Hausman tests. Furthermore, the year dummy variable is added to investigate whether there is an individual time effect. The result shows that the null hypothesis of “no time effect” is rejected, so the two-way fixed effect model is finally selected. The model set in this paper is as follows:(2)$$Tsdea_{it}=\alpha +\beta_{1}VFI_{it}+\beta X_{it}+\mu_{i}+\lambda_{t}+\varepsilon_{it}$$

$Tsdea_{it}$ is the dependent variable representing the fiscal health expenditure efficiency of province *i* in year *t*, which has been specifically introduced above. $VFI_{it}$ is the core independent variable vertical fiscal imbalance, $X_{it}$ is the control variable, including official promotion pressure, per capita GDP, aging level, urban-rural income gap, population density and education level. $\mu_{i}$ is fixed effect of province, $\lambda_{t}$ is fixed effect of time, and $\varepsilon_{it}$ is random error term.

### 5.2. Variable Definition

#### 5.2.1. Core Independent Variable

Vertical fiscal imbalance..Based on traditional methods, this paper fully considers China’s fiscal decentralization system, and uses the methods of Eyraud, Lusinyan (2013) for measurement. That is, vertical fiscal imbalance = 1 − (revenue decentralization/expenditure decentralization)/(1 − fiscal deficit ratio) [30,32]. The revenue decentralization is the per capita fiscal revenue within the provincial budget/(the per capita fiscal revenue within the provincial budget + the per capita central fiscal revenue within the budget). Expenditure decentralization is the per capita fiscal expenditure within the provincial budget/(per capita fiscal expenditure within the provincial budget + the per capita central fiscal expenditure within the budget). As shown in Figure 2, the degree of vertical fiscal imbalance has shown a downward trend of fluctuation, with a temporary rise in the middle time.

#### 5.2.2. Control Variables

Official promotion pressure. In this paper, the official promotion pressure index mainly measures the competitive pressure under the mode of “economic championship”, and comprehensively reflects the promotion competition degree of local governments by using indicators such as local real GDP growth rate, urban registered unemployment rate and fiscal surplus [67]. The specific assignment method of this variable is as follows: the weighted average level of all indicators in all regions is compared with corresponding indicators in this region. When GDP growth rate and fiscal surplus are less than the weighted average level of the year, the value is assigned as 1; otherwise, it is 0. The value is 1 when the unemployment rate is higher than the annual average level; otherwise, it is 0. The scores were then added together to create a local government promotion competition index. The larger the value is, the greater the pressure on the local government for promotion competition is. As a result, local government has a stronger incentive to invest in capital construction expenditure with the characteristics of short investment cycle and fast return, and reduce the investment in health care and other people’s livelihood with long investment cycle, eventually reducing fiscal health expenditure scale efficiency [68]. In addition, under the pressure of promotion, local governments may be more inclined to tilt medical resources to hospitals with high returns, aggravating the imbalance between primary medical institutions and hospitals, and reducing the structure efficiency of fiscal health expenditure.

GDP per capita. The article uses 2006 as the base period to calculate the actual GDP of each province, divide it by the total population of each province to get the real GDP per capita, and finally take the logarithm of the real GDP per capita to get the final control variable Lnpgdp. Economic development provides a strong guarantee for fiscal expenditure, which is conducive to improving the conditions of health resources and improving fiscal health expenditure efficiency. However, large local hospitals with a high level of economic development often have a high reputation, which can easily lead to overcrowding and run on medical resources and may reduce fiscal health expenditure efficiency.

Aging level Aging. The article refers to the proportion of people over 65 years old in the total population of each province. Elderly people generally have more demand for medical treatment, which can fully improve the utilization rate of medical resources, and then improve fiscal health expenditure efficiency. However, it is also easy to lead to excessive medical treatment and waste of medical resources.

Urban-rural income gap Urgap, a ratio of urban per capita disposable income to rural per capita net income in each province. Urban-rural income gap will limit the ability of rural low-income groups to obtain health resources, restrict the expression of interest demands, aggravate the imbalance in the urban and rural health resources utilization, and reduce fiscal health expenditure structure efficiency [69].

Population density. The resident population at the end of each province is divided by the land area of each province, and the logarithm is taken to obtain the final control variable Lnpden. The higher the population density is, the closer the distribution of public health service resources in the province is, and the more significant the scale economy effect of government health resources will be, thus improving fiscal health expenditure efficiency [70].

Education level is measured by the illiteracy rate Illi, the higher the illiteracy rate is, the lower the education level in the area is. The illiteracy rate is equal to the proportion of illiterate people to the population aged 15 and over. Education can improve the supervision ability of local residents to local government, improve the governance level of local government, narrow the gap in the utilization of medical resources, and promote fiscal health expenditure efficiency [71,72].

### 5.3. Data Sources and Statistical Characteristics

The data in this paper cover 31 provinces in China. The input-output index of fiscal health expenditure efficiency mainly comes from China’s Health Statistical Yearbook, which is an annual data publication reflecting the development of health undertakings in China and the health status of residents. China’s medical and health system includes hospitals, primary medical and health institutions (community health service centers, township health centers, village clinics, etc.) and professional public health institutions (CDC, maternal and child health care centers, specialized disease containment centers, etc.). Because the data before 2009 did not include the health services of all types of medical institutions in each province, only the health services of hospitals and some primary medical and health institutions in each province were counted, which led to the fact that the statistical data before and after may be of different caliber. Therefore, this paper Select 2009 and later data for empirical analysis. The data of vertical fiscal imbalance and per capita transfer payment mainly come from China Financial Yearbook. The three indicators of official promotion pressure are mainly derived from China Statistical Yearbook and China Financial Yearbook from 2009 to 2019. Data on aging level, income gap between urban and rural areas, population density and illiteracy rate all come from China Financial Yearbook. The statistical characteristics of all variables are shown in Table 2.

## 6. Results and Discussion

### 6.1. Basic Empirical Results Analysis

The basic empirical results of the model are shown in Table 3, column (1)–(7) in Table 3 report the results of stepwise regression analysis. When no control variable is added in column (1), vertical fiscal imbalance has a positive impact on fiscal health expenditure efficiency at the level of 5%. Columns (2)–(7) gradually add control variables, and the vertical fiscal imbalance still has a positive impact on the fiscal health expenditure efficiency at the level of 5%. According to the theoretical analysis of this paper, there are positive and negative mechanisms of vertical fiscal imbalance on fiscal health expenditure efficiency. The above empirical estimation results show that the current vertical fiscal imbalance in China mainly affects the efficiency of local government health expenditure through a positive mechanism, or that the positive mechanism is more effective than the negative mechanism. Moderate vertical fiscal imbalance can correct the local government’s tendency to focus on capital construction expenditure, regulate the behavior of local governments, ensure the proportion of medical expenditure to a certain extent, and narrow the gap in the allocation of medical resources, which is conducive to the improvement of fiscal health expenditure efficiency. Moreover, the fiscal pressure under the moderate level of vertical fiscal imbalance also promotes the improvement of fiscal health expenditure efficiency.

The pressure of official promotion has a negative impact on the fiscal health expenditure efficiency, and it always remains significant at the level of 1% in the stepwise regression of column (2)–(7).This may be because the pressure of official promotion distorts the public expenditure structure of local governments, squeezes health expenditure, aggravates the structural imbalance of health resources, and damages the fiscal health expenditure efficiency. The aging level and population density have a positive impact on fiscal health expenditure efficiency, and are significant at the level of 5% and 1%, respectively. This is because the more the aging population or population density is, the greater the medical care demand is, thus contributing to the full utilization of health resources and the improvement of fiscal health expenditure efficiency. The urban income gap and illiteracy rate have a negative impact on fiscal health expenditure efficiency, which means that the gap in the use of medical resources among residents will also increase, thereby reducing the fiscal health expenditure efficiency, although the impact is not significant. GDP per capita has a negative impact on the fiscal health expenditure efficiency, but the impact is not significant. It may be because it has both positive and negative effects on fiscal health expenditure efficiency, which cancel each other out.

### 6.2. Robustness Test

#### 6.2.1. Independent Variable and Dependent Variable Change

In order to verify the robustness and credibility of the empirical results, the independent variable and dependent variable were replaced respectively for robustness analysis. First, Malmquist index method is used to calculate TFP, which reflects the efficiency change compared with the previous year [73]. In order to reflect the change over a long period of time, TFP is converted by cumulative multiplication based on 2009, and is used to measure the dependent variable for robust regression analysis. The column (1) shows that the vertical fiscal imbalance has a significant positive influence on malmquist index at the level of 5%. In addition, this paper still follows the traditional method and draws on Jia Junxue’s method, namely “VFI2 = 1 − own income/own expenditure”, to re-measure the vertical fiscal imbalance and conduct robustness analysis [31]. The government’s own revenue and expenditure refer to the general public budget revenue and expenditure of the local government. The empirical results are shown in column (2) of Table 4. Vertical fiscal imbalance has a significantly positive effect on fiscal health expenditure efficiency at the level of 10%, which once again proves the robustness of the empirical results.

#### 6.2.2. Value Healthcare

The ultimate goal of medical and health services is to improve residents’ health, that is to say, the real meaning of improving fiscal health expenditure efficiency lies in the improvement of residents’ health level. Due to data limitations, output indicators that can directly reflect residents’ health outcomes are very scarce, such as emergency mortality and incidence of class A and B infectious diseases. Most of them are indicators of medical service utilization, such as outpatient visits and hospitalization numbers. There are concerns that fiscal health expenditure efficiency cannot truly reflect the quality of medical service. Therefore, this paper comprehensively uses health indicators such as residents’ self-rated health, and chronic disease in charls 2011, 2013, 2015 and 2018 data to investigate whether fiscal health expenditure efficiency really reflects the quality of medical services. In addition, this paper also focuses on whether the cost burden will increase when the fiscal health expenditure efficiency is improved and residents’ medical utilization is improved. Therefore, this paper also adopts charls database to calculate the total medical out-of-pocket expenses of residents and chooses two-part model after a series of tests. As shown in Table 5, the dependent variable is the fiscal health expenditure efficiency, the independent variables in columns (1) and (2) are “self-rated health” and “number of chronic diseases” respectively, and (3) and (4) are the regression results of the two-part model.Control variables at the individual level and city level are also added to the regression, and the individual effect and time effect are controlled. The results show that fiscal health expenditure efficiency is significantly positively correlated with residents’ self-rated health level and the probability of residents’ total out-of-pocket expenses respectively at the level of 5% and 1% but had no significant relationship with the number of chronic diseases and the total out-of-pocket expenses. This indicates that the fiscal health expenditure efficiency calculated in this paper can better reflect the quality of medical services and is not accompanied by the increase in residents’ medical burden, so the efficiency calculation in this paper is relatively scientific.

### 6.3. Mechanism Analysis

Both the size and structural configuration of government health spending can affect its efficiency. Appropriate scale of health resources can exert the scale effect and improve fiscal health expenditure efficiency. Similarly, a reasonable structural allocation of health resources can help balance the medical utilization of different groups and alleviate the problem of “difficult and expensive medical treatment”. This paper mainly studies the resources allocation between primary medical institutions and hospitals, which has an important impact on the medical utilization gap. As shown in Table 6, this paper establishes a mediation effect model to test how the government, in terms of scale and structure, takes measures to improve fiscal health expenditure efficiency in the context of vertical fiscal imbalance. The test formula is as follows:(3)$$\begin{array}{l} Health_{it}=\chi_{0}+\chi_{1}VFI_{it}+\chi_{2}X_{it}+\eta_{it}\\ Tsdea_{it}=\delta_{0}+\delta_{1}Health_{it}+\delta_{2}VFI_{it}+\delta_{3}X_{it}+\eta_{it} \end{array}$$

$Health_{it}$ is an intermediate variable, representing the scale and structural configuration of health resources. Since fiscal health expenditure between primary medical institutions and hospitals is not available, this paper uses the number of beds in health institutions and the number of health technicians to measure health resources. This paper, respectively, adopts the number of health technicians per thousand people Ppers and the number of beds in health institutions per thousand people Pbed to measure the scale of health resources, and respectively uses the number of health technicians per thousand people in primary medical institutions/(number of health technicians per thousand people in primary medical institutions + number of health technicians per thousand people in hospitals) Ppgap, and the number of beds per thousand people in primary medical institutions/(the number of beds in primary medical institutions per thousand people + number of beds per thousand people in hospitals) Pbgap to measure the structural allocation of health resources between different levels of medical institutions. The empirical results are shown in Table 6. In the first stage (1)–(4) regression, China’s vertical fiscal imbalance has a significantly positive impact on Pbed, Ppgap and Pbgap, but has no significant impact on Ppers. In the second-stage regression, as shown in columns (5)–(8), the vertical fiscal imbalance and the scale and structure allocation variables of health resources are added at the same time, and only Pbgap has a significant effect on fiscal health expenditure efficiency at the level of 1% with a coefficient of 1.615, and the significance level of vertical fiscal imbalance also reduce to 10%.This shows that vertical fiscal imbalance can improve fiscal health expenditure efficiency by narrowing the bed resources gap between primary medical institutions and hospitals. The table also lists the effect of the control variable official promotion pressure on health resources. In the first step of regression, it has a significant negative effect on both Ppgap and Pbgap. In the second step of the regression, the Ppgap has no significant effect, and Pbgap has a significant negative effect on fiscal health expenditure efficiency. This indicates that the pressure of official promotion aggravates the imbalance of bed resource allocation between primary medical institutions and hospitals, and then reduces fiscal health expenditure efficiency.

### 6.4. Heterogeneity Analysis

#### 6.4.1. Heterogeneous Fiscal Health Expenditure Efficiency

The baseline regression results show that vertical fiscal imbalance has an overall significant positive effect on fiscal health expenditure efficiency at the present stage. Then, should vertical fiscal imbalance be continuously enhanced to improve fiscal health expenditure efficiency? Therefore, this paper adopts the quantile regression method to test the policy effect of vertical fiscal imbalance at different fiscal health expenditure efficiency levels and selects 10 typical quantiles for investigation. The results are shown in Table 7. It can be found that the impact of vertical fiscal imbalance on fiscal health expenditure efficiency gradually changes from positive to negative with the improvement of efficiency. It has a significantly positive impact at loci 0.1 and 0.4 (low efficiency area), and a significantly negative impact at loci 0.7 and 0.8 (high efficiency area). This indicates that vertical fiscal imbalance does not always improve fiscal health expenditure efficiency, and the degree of vertical fiscal imbalance should not be too high, especially in high-efficiency areas. This is because areas with high efficiency are often accompanied by a high level of vertical fiscal imbalance, which will be excessive if it continues to increase, thus reducing fiscal health expenditure efficiency. Areas with low efficiency are often accompanied by a low level of vertical fiscal imbalance, which can be increased within a moderate range to improve fiscal health expenditure efficiency.

#### 6.4.2. Heterogeneous Transfer Payment Level

As pointed out above, under China’s fiscal decentralization system, a certain degree of vertical fiscal imbalance is often accompanied by the corresponding transfer payment level, which has also been confirmed by the research of Chu Deyin et al. [74]. Moderate central transfer payment can correct the behavior of local governments. For example, by specifying the use of funds, local governments can be guided to increase the expenditure of health and other livelihood projects and adjust the structural allocation of health resources, thus affecting the effect of vertical fiscal imbalance on fiscal health expenditure efficiency. Excessive transfer payments have the opposite effect. Therefore, this paper adopts per capita transfer payment to conduct heterogeneity analysis and observe the heterogeneity of the impact of vertical fiscal imbalance on fiscal health expenditure efficiency under different transfer payment levels. The calculation process of per capita transfer payment is as follows: first, the net transfer payment is calculated by subtracting the general budgetary income from the general budgetary expenditure of each province, and then divided by the total population of each province to obtain the per capita transfer payment, and the natural logarithm is taken to obtain the final control variable Ptrans.

The empirical results are shown in Table 8. The interaction term between vertical fiscal imbalance and per capita transfer payment has a significant positive impact on fiscal health expenditure efficiency at the level of 1%, while vertical fiscal imbalance has a significant negative impact on fiscal health expenditure efficiency at the level of 1%. On the one hand, this indicates that the current transfer payment mainly plays a role in correcting the effect of vertical fiscal imbalance. Additionally, the higher the transfer payment is, the higher the fiscal health expenditure efficiency is. On the other hand, compared with the positive effect of vertical fiscal imbalance in the previous baseline regression analysis, the effect of vertical fiscal imbalance on fiscal health expenditure efficiency turns to negative here. This may be because vertical fiscal imbalance itself has a direct negative effect on fiscal health expenditure. But under the correction effect of transfer payment and other mechanisms, the positive effect is greater than the negative effect, and thus the overall effect of vertical fiscal imbalance is positive.

## 7. Conclusions 

In the theoretical part, the paper analyzes two mechanisms of vertical fiscal imbalance on fiscal health expenditure efficiency and holds that there are positive and negative effects. In the empirical part, the paper first uses the super-efficiency DEA model to calculate fiscal health expenditure efficiency. In the second stage, it adopts the fixed effect model to analyze the effect of vertical fiscal imbalance on fiscal health expenditure efficiency. The conclusion shows that it has a overall positive effect, indicating that the positive effect of vertical fiscal imbalance in China is greater than the negative effect. After changing the measurement of vertical fiscal imbalance, the results still remain robust. Further analysis shows that the vertical fiscal imbalance mainly improves fiscal health expenditure efficiency by narrowing the gap between primary medical institutions and hospitals.

However, the heterogeneity analysis finds that with the improvement of the fiscal health expenditure efficiency, the effect of vertical fiscal imbalance gradually changes from positive to negative, which indicates that vertical fiscal imbalance will not always improve fiscal health expenditure efficiency. Moreover, in the heterogeneous analysis, the coefficient of vertical fiscal imbalance becomes significantly negative, and the cross-term coefficient of transfer payment and vertical fiscal imbalance is significantly positive. This shows that vertical fiscal imbalance itself has a direct negative effect on fiscal health expenditure efficiency, but transfer payment has a corrective effect. In addition, the study also finds that one of the control variables, official promotion pressure has a significant negative impact on fiscal health expenditure efficiency, which is mainly achieved by widening the gap between primary medical institutions and hospitals resources.

## 8. Policy Implications

This paper describes the behavior of local government in the field of health expenditure under the background of vertical fiscal imbalance, expanding the research on local government expenditure behavior under the Chinese decentralization system. In addition, the paper also has some policy implications:(1)Carry out the reform of fiscal decentralization in a scientific way. Moderate vertical fiscal imbalance is helpful to improve fiscal health expenditure efficiency. But excessive imbalance will have the opposite effect, especially in high-efficiency areas. Therefore, the degree of vertical fiscal imbalance should not be too high. This paper holds that China’s financial system reform should pay attention to the following aspects: First, we should divide the powers and expenditure responsibilities of central and local governments in a scientific way, maintain an appropriate level of vertical fiscal imbalance, and make full use of the central government in macroeconomic regulation and in stimulating the initiative of local governments. Secondly, build fiscal relations between the central and local governments according to local conditions, adopt differentiated fiscal policies to optimize the structural allocation of local government public expenditure, and promote the sustainable and healthy development of the economy and the balanced and efficient development of people’s livelihood such as health care.(2)Rationalize the allocation of health resources. The unreasonable distribution of medical and health resources will increase the gap of residents’ medical utilization and reduce the fiscal health expenditure efficiency. Local governments should continue to optimize and promote the health reform at the primary level, such as the medical community and family doctor contract services, to provide residents with efficient and high-quality medical and health services. In particular, the COVID-19 pandemic has highlighted the role of primary medical institutions as the first line of defense, and balanced allocation of health resources should be the top priority in future health system construction.(3)Optimize the transfer payment system. As an important policy tool for the central government to make up the gap between local government’s own income and expenditure, moderate transfer payment corrects the negative effect of vertical fiscal imbalance on fiscal health expenditure efficiency. We should optimize the transfer payment system, reasonably design the transfer payment structure, weaken the dependence of local governments on the transfer payment, avoid the adverse effect of the transfer payment distorting the structure of public expenditure, strengthen the incentive effect of the transfer payment, and optimize the structural allocation and resource efficiency of livelihood areas such as medical care.(4)Reasonably improve the evaluation system for officials promotion. We should further reform the “GDP-only” assessment system, comprehensively consider the ecological, cultural, social and other benefits of local governments and give them appropriate assessment weights. Besides, we should improve the performance evaluation weight of the supply efficiency of public products with strong welfare attributes such as medical care and correct the expenditure tendency of local governments to focus on infrastructure construction and less on people’s livelihood.

## Figures and Tables

**Figure 1 ijerph-20-02060-f001:**
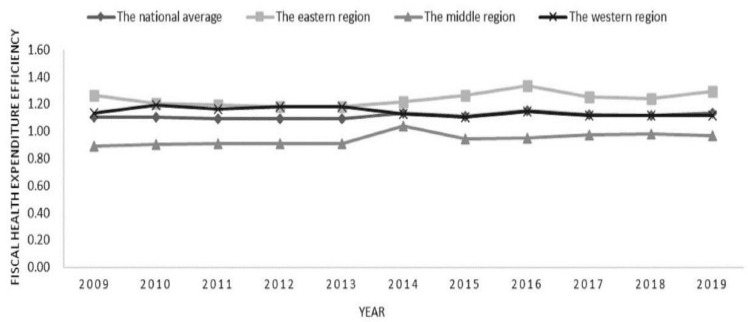
Changes in the fiscal health expenditure efficiency by region in China from 2009 to 2019.

**Figure 2 ijerph-20-02060-f002:**
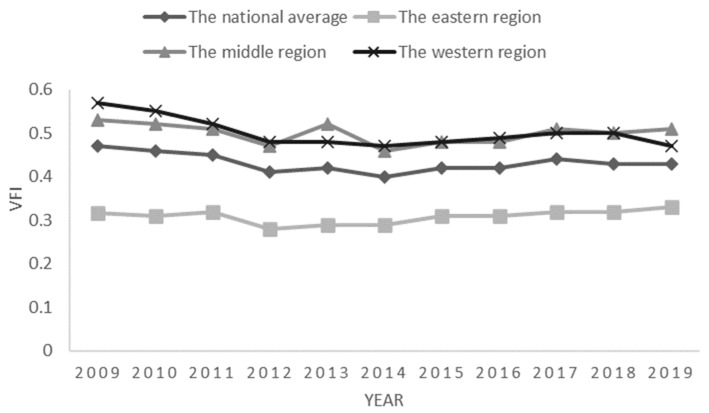
Change of vertical fiscal imbalance in China by region from 2009 to 2019.

**Table 1 ijerph-20-02060-t001:** Fiscal health expenditure efficiency index system.

Variable Types	Variables	Unit
Input indicators	Number of institutions	Piece
	Number of beds	Piece
	Number of health personnel	People
	Number of medical practitioners	People
	Number of Registered nurses	People
Output indicators	Number of outpatient visits	Time
	Number of cases in observation room	People
	Number of health check-ups	People
	Emergency fatality rate (reciprocal)	Percent
	In-patient number	People
	Number of discharged patients	People
	Inpatient operation times	Time
	Prenatal examination rate	Percent
	Incidence of Class A and B infectious diseases (reciprocal)	Percent

**Table 2 ijerph-20-02060-t002:** Description of statistical characteristics of variable.

Variable Name	Observations	Mean	Standard Error	Min	Max
Tsdea	341	1.1132	0.3214	0.65	2.66
VFI	341	0.4316	0.1297	0.1	0.91
VFI2	341	0.5058	0.2064	−0.1	0.95
Pres	341	1.6892	0.8490	0	3
Ptrans	339	8.3762	0.6830	6.8	10.93
Aging	341	0.0992	0.0229	0.05	0.163
Urgap	341	2.7459	0.4851	1.85	4.28
Lnpgdp	341	10.6887	0.4960	9.31	12.01
Lnpden	341	5.3796	1.4506	0.69	8.25
Illi	341	6.1656	6.1523	1.23	41.18
Ppers	248	6.1076	1.3385	3.03	12.44
Pbed	341	4.8023	1.1262	2.56	7.54
Ppgap	248	0.3080	0.0469	0.21	0.6
Pbgap	341	0.21	0.07	0.04	0.37

**Table 3 ijerph-20-02060-t003:** Empirical results of the impact of vertical fiscal imbalance on fiscal health expenditure efficiency in China.

	Dependent Variable (Tsdea)						
	(1)	(2)	(3)	(4)	(5)	(6)	(7)
VFI	0.546 ** (−0.207)	0.653 ** (−0.224)	0.594 ** (−0.203)	0.607 *** (−0.187)	0.555 ** (−0.244)	0.546 ** (−0.241)	0.522 ** (−0.229)
Pres		−0.027 *** (−0.008)	−0.032 *** (−0.006)	−0.031 *** (−0.007)	−0.033 *** (−0.007)	−0.03 *** (−0.009)	−0.029 *** (−0.009)
Aging			3.06 ** (−1.337)	3.167 * (−1.508)	3.185 * (−1.54)	4.006 ** (−1.343)	3.951 ** (−1.411)
Urgap				−0.026 (−0.045)	−0.039 (−0.063)	−0.045 (−0.063)	−0.044 (−0.068)
Lnpgdp					−0.074 (−0.121)	−0.078 (−0.123)	−0.086 (−0.125)
Lnpden						0.097 *** (−0.025)	0.094 *** (−0.026)
Illi							−0.012 (−0.012)
Year FE	Yes	Yes	Yes	Yes	Yes	Yes	Yes
Pro FE	Yes	Yes	Yes	Yes	Yes	Yes	Yes
N	341	341	341	341	341	341	341
Adjusted R^2^	0.0509	0.0626	0.0920	0.0926	0.0944	0.1402	0.1504

Note: *, ** and *** respectively represent significant at the level of 10%, 5% and 1%. The parentheses are standard errors adjusted by Robust. The following table is the same.

**Table 4 ijerph-20-02060-t004:** Empirical results of the impact of vertical fiscal imbalance on fiscal health expenditure efficiency in China (robustness test).

Independent Variable	Dependent Variable	
	(1) Malmquist	(2) Tsdea
VFI	0.095 ** (0.039)	
VFI2		0.282 * (0.142)
Control variables	Yes	Yes
Year FE	Yes	Yes
Pro FE	Yes	Yes
N	330	341
Adjusted R^2^	0.1863	0.1463

Note: *, ** respectively represent significant at the level of 10%, 5%. The parentheses are standard errors adjusted by Robust.

**Table 5 ijerph-20-02060-t005:** Empirical results of the impact of fiscal health expenditure efficiency on resident health in China.

Independent Variable	Dependent Variable			
	(1) Ordered Logit	(2) Poison	(3) Logit	(4) Glm
	Self-Assessed Health	Number of Chronic Diseases	Total Out-of-Pocket Expenses	Total Out-of-Pocket Expenses
Tsdea	1.045 ** (0.447)	−0.061 (0.178)	1.370 *** (0.141)	−0.345 (0.246)
Control variables	Yes	Yes	Yes	Yes
Year FE	Yes	Yes	Yes	Yes
Individual FE	Yes	Yes	Yes	Yes
N	19837	19580	50915	4271
Adjusted R^2^	0.005	0.196		

Note: **, *** respectively represent significant at the level of 5%, 1%. The parentheses are standard errors adjusted by Robust.

**Table 6 ijerph-20-02060-t006:** The path of vertical fiscal imbalance in China to improve fiscal health expenditure efficiency.

Mediated Transmission Mechanism Test								
	The First Step				The Second Step			
Intermediate Inspection Procedure	(1) Ppers	(2) Pbed	(3) Ppgap	(4) Pbgap	(5) Tsdea	(6) Tsdea	(7) Tsdea	(8) Tsdea
VFI	−0.043 (0.123)	0.59 * (0.293)	0.027 ** (0.009)	0.041 * (0.021)	0.337 * (0.163)	0.523 ** (0.225)	0.336 * (0.165)	0.455 * (0.219)
Pres	0.028 (0.025)	0.065 (0.037)	−0.006 ** (0.002)	−0.002 ** (0.001)	−0.031 (0.018)	−0.029 ** (0.01)	−0.032 * (0.016)	−0.026 ** (0.009)
Ppers					−0.026 (0.026)			
Pbed						−0.002 (0.021)		
Ppgap							0.072 (0.253)	
Pbgap								1.615 *** (0.495)
Control variables	Yes	Yes	Yes	Yes	Yes	Yes	Yes	Yes
Year FE	Yes	Yes	Yes	Yes	Yes	Yes	Yes	Yes
Pro FE	Yes	Yes	Yes	Yes	Yes	Yes	Yes	Yes
N	248	341	248	341	248	341	248	341
Adjusted R^2^	0.8941	0.9233	0.2231	0.8112	0.1560	0.1504	0.1540	0.1722

Note: *, ** and *** respectively represent significant at the level of 10%, 5% and 1%. The parentheses are standard errors adjusted by Robust.

**Table 7 ijerph-20-02060-t007:** Quantile regression results.

Quantile	0.1	0.2	0.3	0.4	0.5
VFI	0.3171 ** (0.1312)	0.1337 (0.1586)	0.2303 (0.1458)	0.4009 ** (0.1757)	0.1410 (0.2122)
quantile	0.6	0.7	0.8	0.9	
VFI	−0.1335 (0.1849)	−0.3472 ** (0.1403)	−0.5854 *** (0.1197)	−0.4256 (0.4315)	
Control variables	Yes	Yes	Yes	Yes	Yes
N	341	341	341	341	341

Note: **, *** respectively represent significant at the level of 5%, 1%.

**Table 8 ijerph-20-02060-t008:** Heterogeneity analysis of the impact of vertical fiscal imbalance on fiscal health expenditure efficiency in China.

	(1)
	Tsdea
VFI	−2.005 *** (0.468)
Ptrans	−0.245 (0.204)
VFIPtrans	0.268 *** (0.045)
Control variables	Yes
Year FE	Yes
Pro FE	Yes
N	339
Adjusted R^2^	0.1672

Note: *** respectively represent significant at the level of 1%. The parentheses are standard errors adjusted by Robust.

## Data Availability

The data presented in this study are available upon reasonable request from the corresponding author.

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
