# Peer review of "Effects of Vertical Fiscal Imbalance on Fiscal Health Expenditure Efficiency—Evidence from China"

_ijerph, 2023, doi:10.3390/ijerph20032060_

Round 1

Reviewer 1 Report (Previous Reviewer 2)

In general, the author did a good job of revising the draft. There are still some issues with identification and writing clarity. I would prefer that the author can fix them as much as possible.

BTW. line 626 the number 6 is in a wrong position.

Author Response

Point 1: There are still some issues with identification and writing clarity. I would prefer that the author can fix them as much as possible.

Response 1: Thank you for your valuable advice. In the literature part, the relationship between vertical fiscal imbalance and fiscal health expenditure efficiency is introduced in more detail, and more cohesive sentences are added to improve the fluency of the article.

Point 2:  Line 626 the number 6 is in a wrong position.

Response 2:The error has been corrected accordingly.

Reviewer 2 Report (Previous Reviewer 3)

The authors made careful revision and now this research meets the requirements of publication.

Author Response

Thank you for your review and affirmation.

Reviewer 3 Report (New Reviewer)

Please see the file attached.

Author Response

Point 1:The way the literature in the field and the statistical analysis are presented is confusing and they are not explained clearly enough and on point, even if on the surface the length of the explanations might have you believe that they are sufficient and all-encompassing. The statistical tools that are used are not elaborated on and analyzed to a satisfactory degree.

Response 1: Thank you for your valuable advice.In the literature part, the relationship between vertical fiscal imbalance and fiscal health efficiency is introduced in more detail. And this paper further describes in detail the changing trends of fiscal health efficiency expenditure and vertical fiscal imbalance, and the two variables showed a roughly synchronized trend of change.

Point 2:  I request the authors to comment on the values of R square in the context of the proposed model, especially since its values are sometimes quite low. How significant are the obtained results and to what extent does the proposed model explain these results?

Response 2: Thank you for your valuable advice which makes me have a deeper understanding.The lower R square may be caused by unknown variables or large variation of the data itself.According to the relevant literature, the R square will not deny the importance of any important variables when it is used to study the correlation between relevant variables. Even if the R square is too small, the statistically significant P value still indicates the possibility of these correlation relationships.However, for prediction, the value of R square must reach a certain value, and a low R square can prompt inaccurate prediction value.This paper is mainly used to study the correlation between the explanatory variable and the explained variable,so small R square has little effect on the significance of the coefficient.In addition, the F test in this paper has all passed.And some literature has small R square, such as “Unified medical insurance level of efficiency and fairness * -- Evidence from the integration of medical insurance system for urban and rural residents” published in “economic research”.

Point 3:  Line 502 should specify the fact that (1)- (7) refers to the columns of table 3.The labels of tables 3 and 5 could be clearer so as to better describe the content of these tables. The formatting of the text should be reviewed as certain paragraphs are very confusing (for instance lines 216 - 224, lines 830-831).

Response 3:The error has been corrected accordingly.

This manuscript is a resubmission of an earlier submission. The following is a list of the peer review reports and author responses from that submission.

Round 1

Reviewer 1 Report

The paper is difficult to follow. Aside from requiring a linguistic review, there is a need to explain the concepts that are at the heart of the study such as  "vertical fiscal imbalance" or "fiscal health expenditure efficiency". These may have definitions that are specific to the context, but this context is also not provided. How are resources allocated in China? What are the levels at which there resources allocation decisions are made? How are transfer payments determined? From which instance to which instance? There are key informational elements that are missing that render this study hard to understand.

Although I am familiar with efficiency analysis techniques, I have difficulty understanding this study. The inputs and the outputs are not very clear. There is a list of outputs that are identified but no explanation about why these outputs are selected and very little about how they are measured. Results also seem awkward. They use a DEA but they report an average efficiency over 1. How can this be when efficiency in a DEA is determined by the data and that DMU's efficiency is estimated in proportion to the best DMUs?

Reviewer 2 Report

The effects of vertical fiscal imbalance on China's fiscal health and expenditure efficiency are examined in this paper. Also covered is the impact of vertical fiscal imbalance on expenditure effectiveness. The DEA model is used to calculate the indicator of Effects of Vertical Fiscal Imbalance and to assess the effectiveness of expenditures for fiscal health. According to an empirical study using a fixed-effects model (or, TWFE), China's vertical fiscal imbalance has a generally favorable and significant impact on the effectiveness of fiscal health spending. Further analysis reveals that the size of transfer payments has a corrective effect on the impact of vertical fiscal imbalance and that the impact of vertical fiscal imbalance is changing.

This article's content is in-depth, current, and the author has conducted a significant amount of empirical research. Writing still has a lot of room for development. There are many models, specialized concept terms, and complex empirical designs in the method section, but the introduction is frequently skimpy, and the reader might have trouble understanding it. Additionally, the robustness test of the results is only mediocrely adequate.

Some major concerns

1. This article needs to be written more clearly. The main research question and paper's content should be described in a separate paragraph in the introduction section, and the paper's primary variables should be made clear and introduced early on. The goal is for readers to be able to quickly understand this article's content after reading the introduction.

2. It would appear that the discussion of the mechanism in the second part should be pushed back a little bit, and the current placement is better suited for providing some background.

3. TWFE should be still mainly considered to be under the framework of DID, but the discussion of parallel trend and other assumptions here seems to be not seen in the text.

4. Methods section should also include some discussion on identification threats.

Some minor concerns

1. Under the current approach, standard deviation is typically thought to be clustered at the province level.

2. The picture of Equation 1 needs to be optimized visually, and the labels of the equations seem to be labeled separately for each equation.

3. The percent sign unit in Table 1 is better to expressed in language

4. There is no period at the end of line 157, and there is an extra space at the end of line 175

Reviewer 3 Report

Authors systematically sorts out the multiple theoretical mechanisms of the positive and negative relationship between vertical fiscal imbalance and fiscal health expenditure efficiency. Authors use the super-efficiency DEA model to measure fiscal health expenditure efficiency. Authors adopt the fixed effect model to 509 analyze the effect of vertical fiscal imbalance on fiscal health expenditure efficiency.

Comments

1.In page 6, authors indicate that the national average fiscal health expenditure efficiency showed a fluctuating trend of decline, and increased significantly in 2014 and 2016. But it’s not clear in the Figure 1, I can’t see the fluctuating trend of decline.

2.There are already plenty literatures about fiscal health expenditure efficiency that using the method of DEA and Malmquist exponent. I would recommend to do a robustness test with three-stage DEA (for example, “Measuring green total factor productivity of China’s agricultural sector: A three-stage SBM-DEA model with non-point source pollution and CO2 emissions” ) and Malmquist index. SO we can see the advantage of using a super-efficiency DEA.

3. The contributions of this paper need to be improved.